

# Three distinct classes of myenteric ganglia in mice and humans: insights from quantitative analyses

Luyao Wu[1,2], Lei Xiang[1,2], Yingjian Chen[1,2], Handan Mao[1,2], Xinyao Meng[1,2], Jing Wang[1,2], Honglin Li[1,3], Xuyong Chen[1,2], Jiexiong Feng[1,2] and Jun Xiao[1,2]

[1] Department of Pediatric Surgery, Tongji Hospital, Tongji Medical College, Huazhong University of Science and Technology, Wuhan, China

[2] Hubei Clinical Center of Hirschsprung's disease and allied disorders, Wuhan, China

[3] Nursing Department, Tongji Hospital, Tongji Medical College, Huazhong University of Science and Technology, Wuhan, China

## ABSTRACT

**Background**. The myenteric plexus primarily consists of the myenteric ganglia, which include enteric neurons, synaptic neuropils, and glial cells. Abnormal myenteric plexus formation can result in gastrointestinal disorders. Comprehensive morphological classification studies of myenteric ganglia remain limited.

**Methods**. Whole-mount immunofluorescence staining was used to label myenteric ganglia in colon tissue of mice and children. The ganglionic area and the number of intraganglion neurons were quantified by the K-means clustering algorithm. The guts of embryonic day 11.5 (E11.5) mouse were cultured and immunostained to observe the characteristics of developing myenteric ganglia.

**Results**. Myenteric ganglia can be categorized into three groups in the colon tissues of mice and normal children. A similar classification was observed for Tuj1-positive neuronal cell clusters in the midgut of E11.5 mouse. Culture of the E11.5 mouse midgut revealed that the area of post-cultured clusters of developing neurons also fell into three distinct categories, with a noticeable increase compared to pre-culture.

**Conclusions**. The myenteric ganglia in mice and humans can be categorized into three groups based on both the ganglionic area and intraganglion neuron count, and distinct classes of myenteric ganglia may be present during early development.

Corresponding authors
Jiexiong Feng,
fengjiexiong@tjh.tjmu.edu.cn
Jun Xiao, 13212778286@163.com

## INTRODUCTION

The myenteric plexus (also known as Auerbach's plexus) primarily consists of myenteric ganglia, which include enteric neurons, synaptic neuropils, and glial cells (*Saffrey, 1996*). Positioning between the outer longitudinal and inner circular smooth muscle layers of the intestines provides motor innervation to both muscle layers of the gut and receives inputs from the parasympathetic and sympathetic nervous systems (*Shahrestani & Das, 2024*). Abnormalities in the myenteric ganglia (or plexus) are associated with several intestinal disorders. For instance, hypoganglionosis is defined as a reduction in the ganglion area to less than half of the normal size (*Dingemann & Puri, 2010*). In contrast, hyperganglionosis

(Type B) is characterized by an increased number of intraganglion neurons, which may result in the formation of giant ganglia with an area thrice the size of normal ganglia (*Kapur et al., 2021*; *Muto et al., 2018*).

The myenteric plexus is organized as a mesh network, with each node functioning as a ganglion (*Chevalier et al., 2021*; *Nestor-Kalinoski et al., 2022*). Each ganglion contains a variety of neuronal types, and it is challenging to identify neuronal subtypes, their proportions, morphological characteristics, and functions across different intestinal segments (*Morarach et al., 2021*). Several studies have investigated the morphology of myenteric ganglia in both animal models and humans, focusing on neuronal density, distribution, and subtypes (*Gabella & Trigg, 1984*; *Genov et al., 2023*; *Ippolito et al., 2009*; *Iwase et al., 2005*; *Wattchow, Brookes & Costa, 1995*). Recent advancements include single-cell transcriptome sequencing of enteric neurons and intestinal tissues. However, this approach fails to reconstruct or align original ganglionic structures (*Drokhlyansky et al., 2020*; *Morarach et al., 2021*). Alternatively, studies combining Codex spatial technology with single-cell transcriptome sequencing have provided insights into the spatial structure of intestinal neurons. However, this method merely offers a cross-sectional view of the ganglia rather than capturing their entirety (*Drokhlyansky et al., 2020*).

Given the complexity of neuronal chemical coding, functional types, morphological characteristics, and the size and distribution of ganglia (*Drokhlyansky et al., 2020*; *Morarach et al., 2021*; *Nestor-Kalinoski et al., 2022*), we hypothesized that differences might exist between different ganglia. Therefore, the present study aimed to explore the morphological features of ganglion units by quantifying the area of the myenteric ganglia and the number of intra-ganglionic neurons in mice and humans. Investigating the morphological characteristics of ganglia contributes to understanding their functional traits and provides a new perspective for analyzing their spatial characteristics and interactions with the surrounding smooth muscle.

## MATERIALS & METHODS

### Mice specimens and study approval

Wild-type C57BL/6 mice were purchased from Beijing Vital River Laboratory Animal Technology Co. Ltd. and housed in a specific pathogen-free (SPF)-grade facility at a controlled temperature of 20–25 degrees Celsius (°C) with a 12/12-hour light-dark cycle, and provided with ad libitum food, water, and fresh bedding. These 4-week-old mice were anesthetized with 5% isoflurane, placed in a supine position, and mid-colon tissue was collected following abdominal incision. Mice were then euthanized by intraperitoneal injection of pentobarbital (50 mg/kg) and cervical dislocation. Female mice were bred with males, and the date of fertilization was recorded. Pregnant mice at embryonic day 11.5 (E11.5) and E14.5 were anesthetized with 5% isoflurane, and the uterus was dissected to collect embryos. Embryos were euthanized by decapitation, and the gastrointestinal tract was isolated under a stereomicroscope. Breeding was strictly controlled to meet experimental needs (three biological replicates) while minimizing animal use. At the conclusion of the experiment, any surviving animals were euthanized by intraperitoneal injection of pentobarbital followed by cervical dislocation. All animal procedures were

approved by the Institutional Animal Care and Use Committee of Tongji Medical College, Huazhong University of Science and Technology (Approval number: 2019-S2500). This study was also approved by the Ethics Committee of Tongji Medical College, Huazhong University of Science and Technology (Approval number: 2021-S033). Written informed consent was obtained from all patients or their legal guardians involved in this study.

## Whole-mount immunofluorescence (IF) staining of mouse gastrointestinal tract

The embryonic gut was carefully isolated from E11.5 mice. Colonic tissue samples were obtained from the middle colon of 4-week-old mice, with the mucosal and sub-mucosal layer torn off. The tissue was fixed with 4% paraformaldehyde (PFA) for 1 h at 25 °C, after which it was washed thrice with phosphate-buffered saline (PBS). Prior to the application of the primary antibody, the tissue was incubated with a blocking solution (1% bovine serum albumin (BSA), 0.15% glycine, and 0.1% Triton X-100 in PBS) for 2 h at 25 °C. The primary antibody (Tuj1, mouse, 1:500, Cat# ab78078, Abcam; Pgp9.5, rabbit, 1:500, Cat# 108986, Abcam; HuC/D, rabbit, 1:400, Cat# ab184267, Abcam, Sox10, rabbit, 1:500, Cat# ab155279, Abcam; Sox10, mouse, 1:40, Cat# sc-365692, Santa Cruz; Tac1, rabbit, 1:300, Cat# 13839-1-AP, Proteintech) and secondary antibody (Rhodamine-conjugated goat anti-mouse, 1:50, Cat# SA00007-1, Proteintech, USA; CoraLite488-conjugated goat anti-rabbit, 1:400, Cat# SA00013-2, Proteintech; Rhodamine-conjugated goat anti-rabbit, 1:50, Cat# SA00007-2, Proteintech, USA; CoraLite488-conjugated goat anti-mouse, 1:300, Cat# SA00013-1, Proteintech) were incubated at 4 °C overnight. The devices were stained with anti-fade 4′,6-diamidino-2-phenylindole (DAPI) Fluoromount-G (Cat# 0100-20, Southernbiotech, USA), followed by washing in PBS. Images were captured using a laser confocal microscope (Zeiss LSM800, Germany). Tuj1, a neuronal cytoskeletal protein, is used to delineate nerve fibers, thereby outlining the myenteric plexus (*Gogolou, Frith & Tsakiridis, 2021*). Pgp9.5 is another useful marker that labels nerve fibers and prominently shows neuronal cell bodies, thus aiding in estimating the number of neurons within a ganglion (*Kang, Fung & Van den Berghe, 2021*). HuC/D also labels the neuron cell body, not just the nucleus (*Bodin et al., 2021*; *Sharkey & Mawe, 2023*). Sox10 marks the nuclei of enteric glial cells and assists in identifying the contours of ganglia (*Sharkey & Mawe, 2023*).

## Whole-mount IF staining of human colonic tissue

Normal transverse colonic tissue samples were collected from children (aged 3 and 4 years) who suffered injuries in traffic accidents, whereas hypo-ganglionic colonic tissue samples were obtained from children (two aged 3 years and one aged 4 years) with hypoganglionosis who required surgical intervention. Blood contained in the tissue was rinsed with normal saline. The outer plasma membrane layer, inner mucosa, and submucosa of the colon were removed under a dissecting microscope to obtain muscularis propria samples (*Drokhlyansky et al., 2020*). The tissue samples were fixed over-night with 4% PFA at 4 °C to maintain the samples in a flat position throughout the process. Following fixation, the samples were washed five times for 5 min each. And then a blocking and permeabilization solution containing 10% fetal bovine serum (FBS), 1% Triton X-100, and 0.02% sodium azide was

added at 4 °C overnight. The primary antibody (Tuj1, mouse, 1:400, Cat# ab78078, Abcam and HuC/D, rabbit, 1:400, Cat# ab184267, Abcam) and corresponding secondary antibody were incubated at 4 °C for two nights. Finally, images were captured using a Zeiss Axio Zoom.V16 large-field fluorescence stereo zoom microscope (Zeiss, Germany) at 7–250× magnification with a numerical aperture (NA) of 0.25. After five days of fluid immersion, the colon tissue separated more easily into anatomical structural layers. Therefore, we attempted to separate the intestinal muscle layers with forceps to obtain either longitudinal muscle-myenteric plexus or circular muscle-myenteric plexus samples or unchanged muscular propria that were imaged. As for the thick tissue area under the field, $Z$-axis scanning with a 3-μm scanning thickness was used.

### Culture *in vitro* of embryonic guts and IF staining

Embryonic guts were isolated from E11.5 mice and placed onto the filter membrane (0.45 μm HABG01300; Millipore). The explants were cultured in Dulbecco's Modified Eagle Medium with Nutrient Mixture F-12 (DMEM/F-12) containing 10% FBS, 100 ng/mL glial cell-derived neurotrophic factor (GDNF, Cat# 450-10; PeproTech), and 1% penicillin and streptomycin at 37 °C in a 5% $CO_2$ incubator. GDNF cytokines induce neural precursor cells migration. The culture method used was based on the previous research (*Hearn et al., 1999*; *Nagy et al., 2020*). After 48 h of culture, the explants and filter membranes were fixed with 4% PFA for 1 h at 25 °C, then washed thrice with PBS. Prior to the incubation of the primary antibody (Tuj1, mouse, 1:500, Cat# ab78078, Abcam) and secondary antibody (CoraLite488-conjugated goat anti-mouse, 1:400, Cat# SA00013-1, Proteintech) at 4 °C overnight, the devices were blocked in PBS (1% BSA, 0.15% glycine, and 0.1% Triton X-100 in PBS) for 2 h at 25 °C. The devices were subsequently stained with anti-fade DAPI Fluoromount-G and washed with PBS. Finally, images were captured using a Leica DMi8 laser confocal microscope (Leica, Germany) at 20× magnification with an NA of 0.7.

### Culture of neurospheres

The gastrointestinal tracts of E14.5 mice were extracted. After cutting the tissues into pieces, they were dissociated using 0.25% trypsin (Cat# G4005-100 mL, Servicebio, China) at 37 °C with gentle pipetting. Next, single-cell suspensions were cultured in DMEM/F-12 containing 1% N2 supplement (Cat# 17502048, Gibco, USA), 1% B27 supplement (Cat# 17504044, Gibco), 10 ng/mL bFGF (Cat# HY-P7004, MCE, USA), 10 ng/mL EGF (Cat# HY-P7109, MCE), 1% glutamate (Cat# 25030081, Gibco), and 1% penicillin and streptomycin with 5% $CO_2$ at 37 °C in a humidified incubator. The medium is half-changed every other day. On the 7th day of the culture period, retinoic acid (1 μM, Cat# ab120728, Abcam) was added for a culture period of 2 d (*Li et al., 2023*).

### IF staining of neurospheres

The use of HNK1 and P75 enabled the identification of enteric neural crest cells (ENCCs), and Tuj1 was used to identify the cell body, axons, and dendrites of enteric neurons (*Nagy & Goldstein, 2017*). Neurospheres suspended from the culture system on the 4th day adhered to a polylysine-coated glass coverslip. The neurospheres were fixed with 4% PFA for 30 min

at 25 °C, after which they were washed thrice with PBS. Primary antibody (HNK1, mouse, 1:100, Cat# sc-81633, Santa Cruz, USA; P75, rabbit, 1:50, Cat# ab52987, Abcam, USA; and Tuj1, mouse, 1:500, Cat# ab78078, Abcam) and corresponding secondary antibodies were incubated sequentially. The devices were stained with anti-fade DAPI Fluoromount-G and washed with PBS. Finally, images were captured using a DM6B fluorescence microscope (Zeiss, Germany).

## Flow cytometry of dissociated neurospheres

The suspended neurospheres, which were cultured for either 3 or 6 d, were gently digested using a solution of 0.25% trypsin combined with mechanical blowing. This process resulted in the formation of a single-cell suspension, which was then filtered through a 38-μm cell filter. After cell counting, the single-cell suspension was stained with direct-labeled antibodies (anti-CD271/P75-PerCP, 1:500, Cat#130-128-624, Miltenyi, and anti-CD57/HNK1-APC, 1:20, Cat#359610, BioLegend). Sample data were collected using a flow cytometer (CytoFLEX, Beckman, USA).

## Statistical and morphological analysis

The statistical outcomes depicted in the bar graph are delineated by the mean values accompanied by their corresponding standard errors of the mean (S.E.M.). To facilitate comparative analysis between the two groups, two-tailed Student's $T$ tests were systematically applied. Ordinary one-way analysis of variance (ANOVA) (Tukey's) and multiple comparison tests were used for three or more groups.

Aggregations of neuronal bodies or nuclei, sometimes aided by Sox10-positive glial nuclei, were used to delineate ganglion boundaries (*Hoff et al., 2008*; *Kang, Fung & Van den Berghe, 2021*). In this study, we defined a ganglion as any network node identified within the Tuj1-positive neuronal network. Notably, even when a network node contained only a single neuron (Fig. S1), we considered it a ganglion for statistical analysis, distinguishing our approach from previous study (*Hoff et al., 2008*). The polygon selection tool in Photoshop (version 2021) was used to outline the ganglion boundaries (*Hoff et al., 2008*), measure pixel dimensions, and convert them to actual area measurements using a calibrated ruler, based on the aggregate neuronal bodies and the emitted nerve fibers. The number of neurons within each ganglion was counted manually based on Pgp9.5-positive or HuC/D-positive cell bodies. Additionally, it is important to note that images were captured from or near the mesenteric side because of the absence of ganglion cells at the location of the hypertrophic longitudinal muscle band.

# RESULTS

## Three categories for myenteric ganglion area and intraganglion neuron count

Colonic samples were prepared by dissecting the mucosal and submucosal layers of 4-week-old mice (*Yarandi et al., 2020*). To quantify the area of myenteric ganglia, immunofluorescence (IF) co-staining against Tuj1 and Sox10 was performed to measure the area of myenteric ganglia (Fig. 1A, Data S1). The K-means clustering method was used

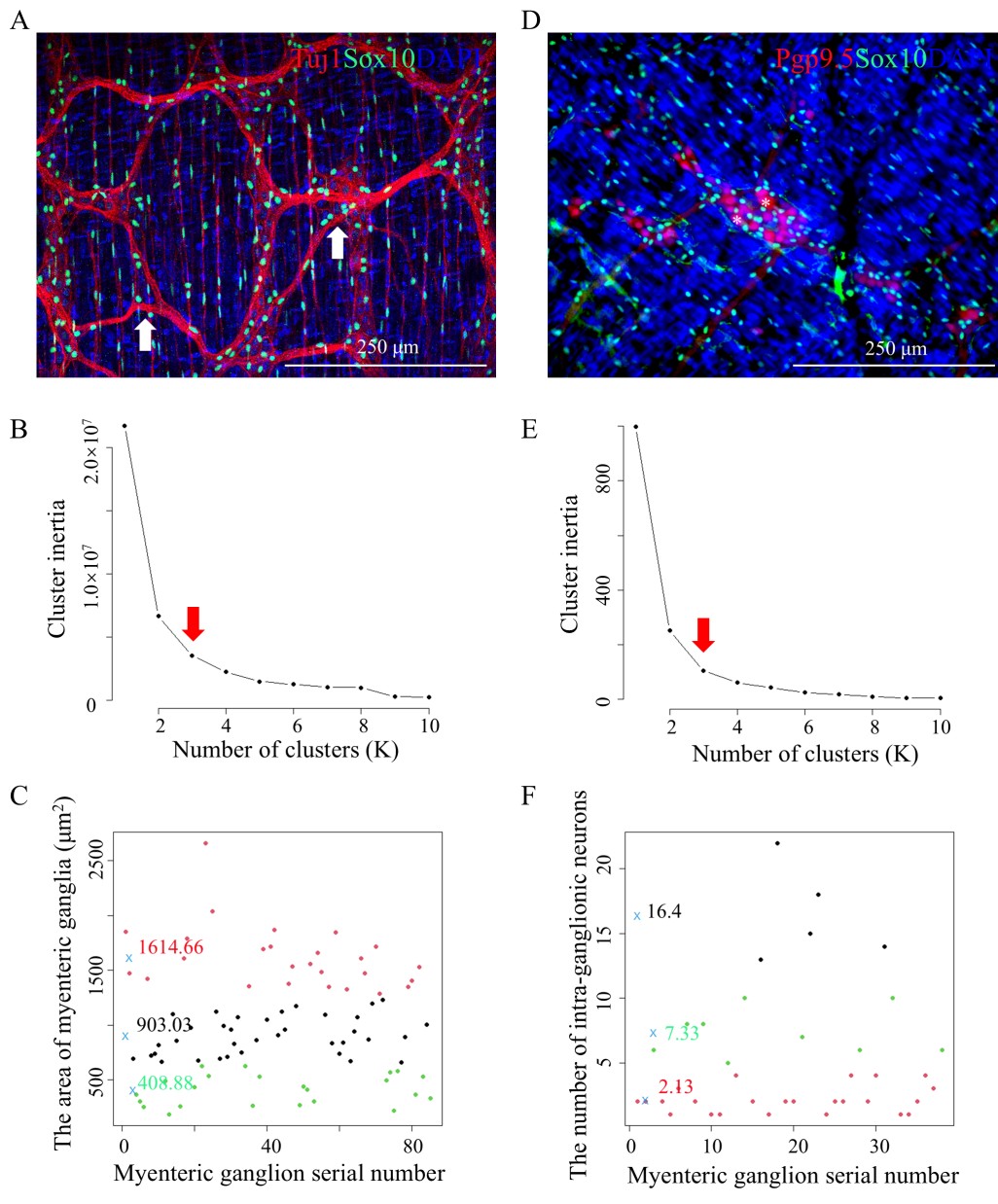

**Figure 1  Three classes of mouse myenteric ganglia area and intra-ganglionic neuron count.** (A) Tuj1 and Sox10 labeled the myenteric plexus in the colons of 4-week-old mice, with white arrows indicating the myenteric ganglion. Four mice were used, eight images were produced, and 86 data of ganglion area were generated. (B) The elbow method was used to determine of the optimal K-value (red arrow) of (A). (C) The classification graph of K-means clustering for (A). (D) Pgp9.5 and Sox10 labeled neurons within the myenteric plexus in the colons of 4-week-old mice. Sox10 marked enteric glial cells, and white asterisks indicated the cell bodies of neurons. Three mice were used, five images were produced, and 38 data of intra-ganglionic neuron count were generated. (E) The elbow method was used to determine of the optimal K-value (red arrow) of (D). (F) The classification graph of K-means clustering for (D). Bule x, central point location.

to classify the data set. The optimal K-values were determined using the elbow method (Fig. 1B). Subsequently, K-means clustering was used to identify three distinct clusters with area of $408.88 \pm 137.34$, $903.03 \pm 171.35$, and $1,614.66 \pm 209.97$ μm² (Fig. 1C, Table S1). These methods are commonly used in data clustering (*Loron et al., 2023*; *Pasin & Gonenc, 2023*) and suggest the existence of three distinct groups of myenteric ganglia based on their areas: small, medium, and large.

Similarly, IF co-staining against Pgp9.5 and Sox10 was performed to quantify the number of intraganglion neurons in colonic samples from 4-week-old mice (Fig. 1D, Data S1). The optimal K-values were determined using the elbow method (Fig. 1E). Subsequently, K-mean clustering showed three distinct clusters with neuron counts of $2.13 \pm 1.03$, $7.33 \pm 1.80$, and $16.40 \pm 3.65$ (Fig. 1F, Table S2). This suggests the existence of three diverse groups of myenteric ganglia differentiated by neuron count: one with fewer neurons, one with an intermediate number, and one with more neurons.

### Three categories of mouse myenteric ganglia

To further explore the classification of myenteric ganglia, IF co-staining against Pgp9.5 and Tuj1 was performed on the colonic samples from 4-week-old mice (Figs. 2A and 2B, Data S2). For each individual ganglion, the area and number of intraganglion neurons were quantified simultaneously, resulting in the corresponding pairs of data. The optimal K-value was determined using the elbow method (Fig. 2C), and subsequent K-means clustering analysis revealed that the data can be classified into three clusters (Fig. 2D). In summary, the colonic myenteric ganglia can be classified into three categories based on both area and neuron count (Table 1). Cluster 1 myenteric ganglia exhibited a smaller area and a lower neuron count, whereas cluster 2 myenteric ganglia showed a moderate area and a neuron count that was neither low nor high. Cluster 3 myenteric ganglia, on the other hand, possessed a larger area and a higher neuron count.

To study sex-specific differences in the statistical model of enteric nervous system (ENS) structure, the sex-segregated data was generated with whole-mount IF co-staining against HuC/D and Tuj1. The results showed the colonic myenteric ganglia can be classified into three categories based on both area and neuron count in female (Figs. 3A–3C) and male (Figs. 3D–3F) mice (Table 2, Data S3). Overall, the total neuron count (Fig. S2A) and total ganglia area (Fig. S2B) did not differ significantly between the sexes. However, when focusing on the medium and large ganglia, both the ganglion area and neuron count were larger in female mice compared to male mice, while no significant differences were observed in the small ganglia (Figs. S2C and S2D).

### Three categories of human myenteric ganglia

To explore whether similar morphological classification of myenteric ganglia exist in human colon tissues, normal colon tissue samples were processed into muscularis propria samples. Whole-mount IF staining was conducted using neuron markers Tuj1 and HuC/D (*Sharkey & Mawe, 2023*). The normal myenteric ganglion displayed a regular polygonal shape and was connected to other ganglia *via* nerve fibers, forming a network (myenteric plexus) (Fig. 4A, Data S4). Each node in the network represented an individual myenteric

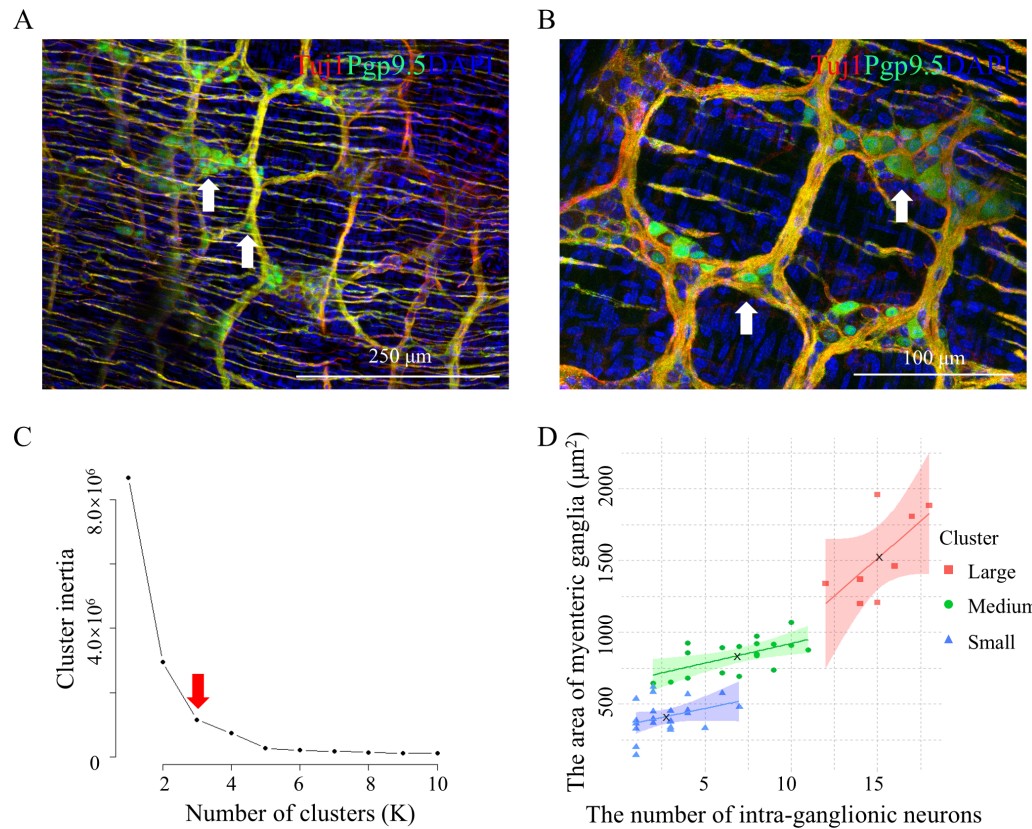

**Figure 2** **Three classes of mouse colonic myenteric ganglia.** (A and B) Tuj1 and Pgp9.5 labeled the ganglia and neurons within the myenteric plexus in the colons of 4-week-old mice, (A) representative image of the overall myenteric plexus network at 200× magnification, (B) representative image of a specific ganglion at 400× magnification; white arrows indicated the myenteric ganglion. Three mice were used, six images were produced, and 49 paired data were generated. (C) The elbow method was used to determine of the optimal K-value (red arrow) of (A and B). (D) The classification graph of K-means clustering for (A and B). Linear regression analysis was performed within each group, resulting in trend lines, and the colored areas represented 95% confidence intervals.

**Table 1** **Information of myenteric ganglionic area and intraganglion neuron count in mice.**

|  | Counts | | Size | | Proportion |
|---|---|---|---|---|---|
|  | M (n) | SD | M (μm²) | SD |  |
| Cluster 1 | 2.77 | 1.69 | 412.68 | 119.23 | 45.83% |
| Cluster 2 | 6.89 | 2.61 | 836.71 | 120.79 | 37.50% |
| Cluster 3 | 15.13 | 1.89 | 1,527.52 | 308.37 | 16.67% |

**Notes.**
M, mean; SD, Standard deviation.

ganglion (Fig. 4B). The ganglionic area and the number of intraganglion neurons were quantified. This analysis revealed that the myenteric ganglia in the human colon can be categorized into three groups (Figs. 4C and 4D, Table 3), similar to those in mice. We performed the same staining procedure on diseased colon specimens obtained from

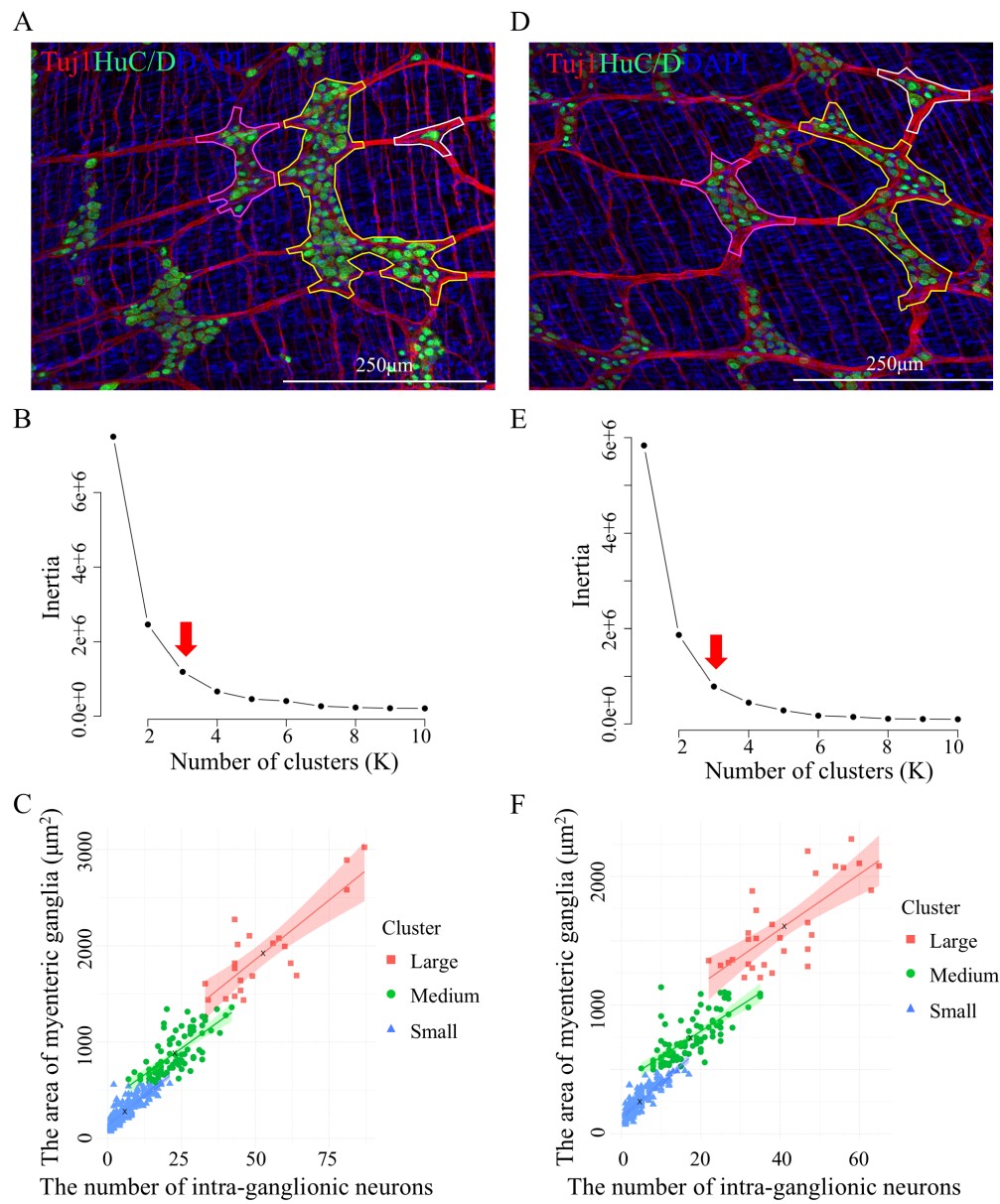

**Figure 3** **Classification of colonic myenteric ganglia into three distinct groups in female and male mice.** (A) Immunofluorescence labeling of colonic myenteric plexus from 4-week-old female mice using Tuj1 and HuC/D to mark ganglia and neuronal cell bodies, respectively. Data were collected from four mice, yielding 16 images and 298 paired data points. The white, magenta, and yellow lines delineate the ranges of Cluster 1, Cluster 2, and Cluster 3 ganglia, respectively. (B) Application of the elbow method to determine the optimal number of clusters (K-value) for the dataset in (A), with the optimal value indicated by the red arrow. (C) K-means clustering classification graph for the female dataset in (A). Linear regression analyses were performed for each cluster, with trend lines and 95% confidence intervals (shaded areas) displayed. (D) Immunofluorescence labeling of colonic myenteric plexus from 4-week-old male mice using Tuj1 and HuC/D. As in (A), the white, magenta, and yellow lines indicate the ranges of Cluster 1, Cluster 2, and Cluster 3 ganglia, respectively. Data were collected from four mice, producing 

**Figure 3 (…continued)**
16 images and 267 paired data points. (E) Application of the elbow method to the male dataset (D) to determine the optimal K-value, indicated by the red arrow. (F) K-means clustering classification graph for the male dataset in (D), with linear regression analyses for each group. Trend lines and 95% confidence intervals are shown, highlighting the distribution patterns across the clusters.

**Table 2 Information of myenteric ganglionic area and intraganglion neuron count in female and male mice.**

| | | Counts | | Size | | Proportion |
|---|---|---|---|---|---|---|
| | | M (n) | SD | M (μm²) | SD | |
| Cluster 1 | Female | 5.76 | 4.92 | 277.88 | 138.34 | 65.77% |
| | Male | 4.60 | 3.52 | 249.16 | 111.44 | 54.68% |
| Cluster 2 | Female | 22.64 | 7.39 | 884.54 | 224.02 | 27.18% |
| | Male | 17.44 | 6.40 | 746.26 | 175.8 | 34.08% |
| Cluster 3 | Female | 52.62 | 15.22 | 1,921.54 | 456.54 | 10.61% |
| | Male | 41.07 | 11.73 | 1,613.15 | 335.80 | 11.24% |

Notes.
M, mean; SD, Standard deviation.

**Table 3 Information of myenteric ganglionic area and intraganglion neuron count in human.**

| | Counts | | Size | | Proportion |
|---|---|---|---|---|---|
| | M (n) | SD | M (μm²) | SD | |
| Cluster 1 | 29.21 | 7.66 | 1,801.59 | 393.83 | 41.38% |
| Cluster 2 | 55.58 | 14.62 | 3,113.70 | 523.80 | 32.76% |
| Cluster 3 | 90.47 | 13.61 | 5,085.85 | 675.75 | 25.86% |

Notes.
M, mean; SD, Standard deviation.

children diagnosed with hypoganglionosis. These results indicated that the morphology of the myenteric plexus was discrepant, with the presence of larger and smaller ganglia (Figs. 4E and 4F). In addition, compared with the normal samples, the hypo-ganglionic tissues exhibited a significantly lower number of neurons per ganglion, smaller ganglion areas, fewer ganglia per field, and a higher proportion of small ganglia (Fig. S3).

## Classification of Tuj1-positive ganglia area in embryonic guts

The number of enteric ganglia is established early in development with the clusters of developing neurons matching the final fully developed ganglia (*Chevalier et al., 2021*). To observe whether a similar phenomenon exists in developing intestinal ganglia during embryonic development, whole-mount IF staining for Tuj1 was conducted on intact gut (midgut and hindgut) obtained from E11.5 mice (Fig. 5A). The results revealed a Tuj1-positive neural network within the midgut that gradually transitioned into the hindgut devoid of Tuj1 signals (Fig. 5D), consistent with the typical migratory pattern of enteric neural crest cells (ENCCs) in E11.5 mouse embryos (*Nagy & Goldstein, 2017*). At the junction of the ileocecal region, the Tuj1-positive neural network appeared sparser, indicating the recent arrival of ENCCs at this location (Fig. 5E, Data S5). In the oral part of

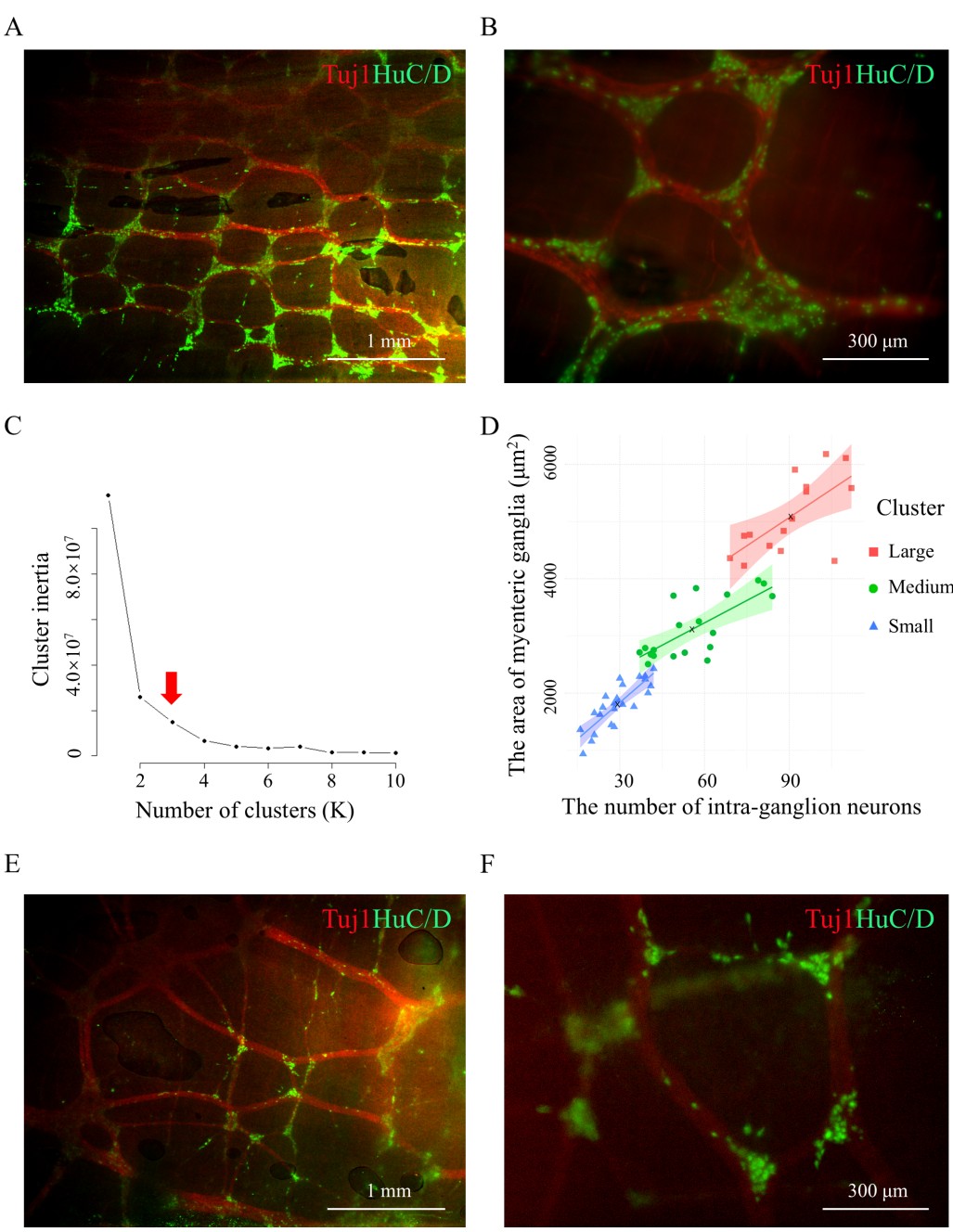

**Figure 4 Classification of human colonic myenteric ganglia.** (A and B) Tuj1 and HuC/D labeled the myenteric plexus and neurons within the muscularis propria of normal colon from two patients. Twenty-four images were produced, and 58 paired data were generated. (C) The elbow method was used to determine of the optimal K-value (red arrow). (D) The classification graph of K-means clustering. (E and F) Tuj1 and HuC/D labeled the myenteric plexus and neurons within the colonic muscularis propria from three patients with hypoganglionosis.

the midgut distal to the ileocecal region, the Tuj1-positive neural network exhibited a more uniform distribution (Fig. 5F, Data S5). Area statistics were performed for Tuj1-positive cell clusters in the midgut. The samples were divided into two groups: anterior section (half of the midgut near the cecum) and posterior section (half of the midgut near the foregut). Using the elbow method and K-means clustering, both the anterior and posterior sections were classified into three distinct clusters (Figs. 5I–5L, Tables S3 and S4). Notably, the area of Tuj1-positive cell clusters in the anterior section was smaller than that in the posterior section (Fig. 5O).

*In vitro* gut culture remains a valuable model for intuitive observation of gut development *in vivo* (Chen et al., 2023; Mandal et al., 2024). Thus, an *in vitro* culture of the E11.5 mouse midgut was conducted to observe the differentiation and migration of neuronal precursors (Figs. 5B and 5C). Tuj1-positive cells on the filter membrane were arranged individually (Fig. 5G), in contrast to the collective Tuj1-positive cell bodies within the intestine, forming neural networks (Fig. 5H, Data S5). Area statistics were obtained for Tuj1-positive cell clusters within the midgut of the filter membrane. Using the elbow method and K-means clustering, these cell clusters were categorized into three distinct groups (Figs. 5M and 5N, Table S5). Notably, the area of these cell clusters was significantly larger than that of the corresponding regions in the posterior sections of the midgut (Fig. 5O).

### Distinct categories for neuronal cells clusters within neurosphere

As aggregates of enteric neurons and glial cells, ganglia are derived from ENCCs during embryonic development (Rao & Gershon, 2018). The *in vitro* growth of enteric neurospheres closely resembles the *in vivo* development of ENCCs into the neural plexus, as neurospheres—comprising of neural stem and progenitor cells—can differentiate into neurons and glial cells (Chen et al., 2023; Mandal et al., 2024). To better understand the formation of ganglia and neural plexus networks, we adopted a mouse primary enteric neurosphere *in vitro* culture model to study the biological behavior of ENCCs (Heumüller-Klug et al., 2023; Schäfer, Hagl & Rauch, 2003). The ENCCs exhibited proliferative capabilities *in vitro* (Fig. S4). IF staining with Tuj1 was performed on neurospheres cultured up to the 7th day, revealing the presence of either none (17.4%), one (31.4%), two (36.3%), or three (14.9%) distinct groups of neuronal cell clusters within the neurosphere (Figs. 6A–6J). Considering the existence of various neuronal cell clusters in isolation or in interconnected networks, it was speculated that distinct categories may exist for the clusters of neuronal cells in the neurosphere.

### DISCUSSION

In this study, we first categorized the myenteric ganglia in mice and humans into three groups based on both the ganglionic area and intraganglion neuron count, which has not been reported previously. Considering that the number of enteric ganglia is constant early in development (Chevalier et al., 2021), the clusters of developing neurons in gut cultures *in vitro* were categorized into three groups and matched to the fully developed ganglia. Furthermore, the classification of Tuj1-positive cell clusters within the neurosphere as a biological model for studying enteric nervous system development reflects the classification

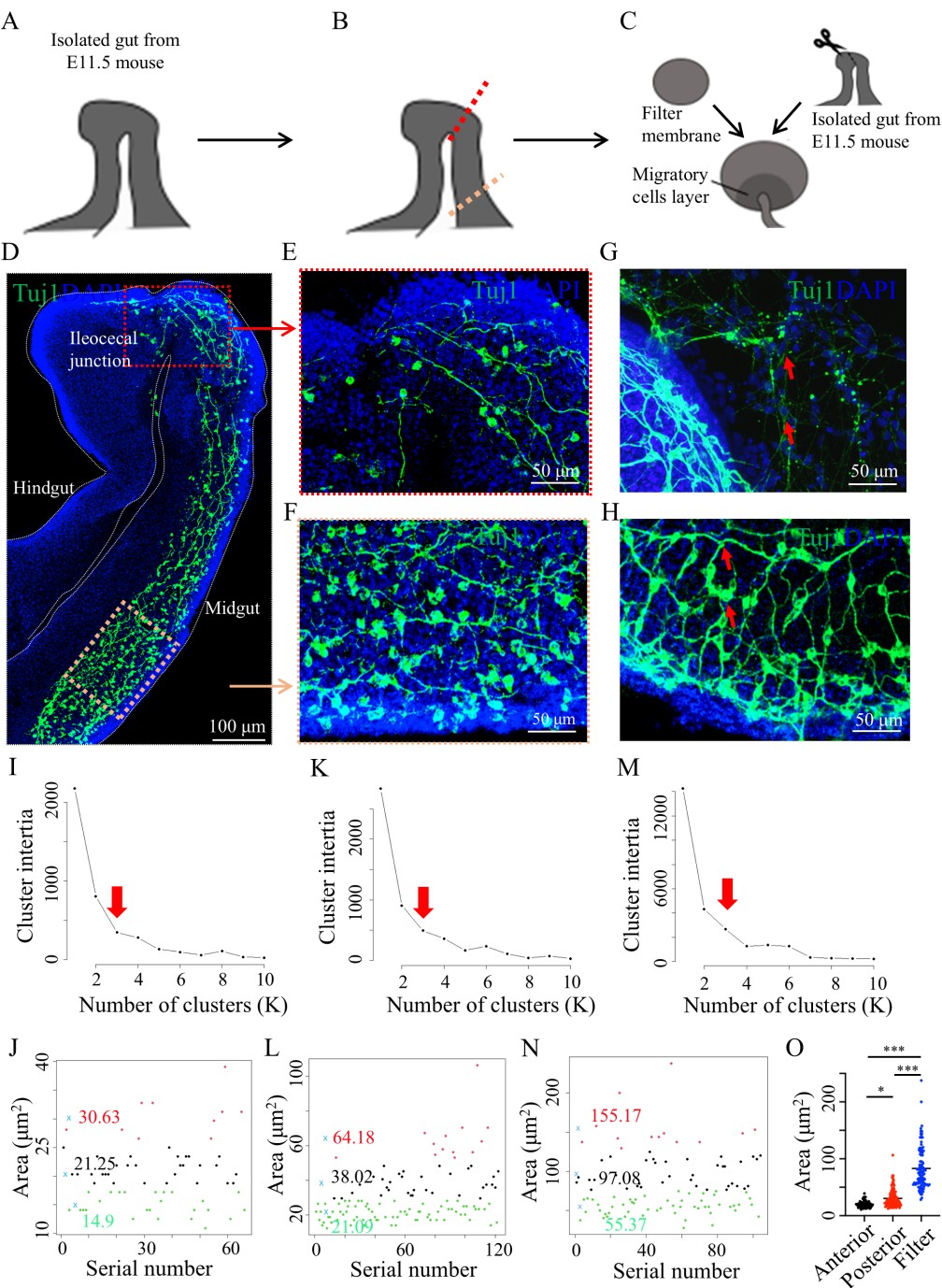

**Figure 5** **Whole-mount staining of embryonic mouse gut and culture *in vitro*.** (A) Schematic diagram of E11.5 mouse gut. (B) The red dashed line indicates the position of migrating Tuj1-positive cells, and the yellow dashed line indicates the location of enteric neuron precursors behind the migrating front Tuj1-positive cells (in front). (C) Schematic diagram of the gut culture *in vitro*. (D) Enteric neuron precursors and neurons were labeled with Tuj1 at E11.5. Three mice were used, five images were produced, and three *(continued on next page...)*

**Figure 5 (…continued)**
sets of area data (66, 122, and 104, respectively) were generated. (E and F) Local magnifications of the red and yellow boxes in (D), respectively. (G) The Tuj1-positive neuron precursors and neurons (red arrows) migrating onto the filter membrane. (H) The clusters of Tuj1-positive cells (red arrows) within the gut. Determination of the optimal K-value (red arrow) for the area of Tuj1-positive cell clusters in the anterior midgut (I), posterior midgut (K), and the midgut on the filter (M) using the elbow method. Classification graph of area of Tuj1-positive cell clusters in the anterior midgut (J), posterior midgut (L), and the midgut on the filter (N) using K-means clustering. (O) Statistical analysis for area of Tuj1-positive cell clusters among the above three groups using one-way ANOVA for multiple comparisons. Anterior, anterior midgut section; posterior, posterior midgut section; filter, posterior midgut section on the filter membrane after culture. *, $p = 0.0200$ ***, $p < 0.001$.

of ganglia in the nerve plexus to a certain extent. These results indicate that there may be distinct classes of myenteric ganglia during early development.

The area of the ganglion and the number of neurons within the ganglion are location-dependent in the gastrointestinal (*Nestor-Kalinoski et al., 2022*). For instance, the number of intra-ganglionic neurons in the proximal colon was higher than that in the middle and distal colons. Additionally, the colonic myenteric ganglion area and neuron count are slightly bigger in female mice than in male mice (*El-Salhy, Sandström & Holmlund, 1999*), consistent with our sex-based data for medium and large ganglia. Furthermore, smaller ganglia are more prevalent than larger ones in the adult murine small intestine (*Kobayashi et al., 2021*). Similarly, our data showed a comparable frequency distribution in mid-colon of 4-week-old mice, with smaller ganglia being more common than larger ones (Fig. S5). We selected the mid-colon of 4-week-old mice as a representative segment to study the intrinsic properties of the enteric ganglia. Whether the ganglionic classification is applicable to the intestinal segments at other locations and ages, requires further investigation. The classification of ganglia may be related to a variety of enteric neuron subtypes. Taking Tac1-positive neurons (the marker of excitatory neuron *Morarach et al., 2021*) as an example, we conducted whole-mount IF assay in male mice to analyze the proportion of Tac1-positive neurons in all neurons among the three clusters of ganglia (Figs. S6A–S6D). The proportion of Tac1-positive neurons in small ganglia was 50% on average, which was higher than that in medium and large ganglia (33.33% and 15.91%). There was no difference between medium and large ganglia (Fig. S6E). The distribution of other neuronal subtypes within the different clusters of ganglia remains to be fully elucidated and further studies are needed.

However, the mechanisms underlying myenteric ganglion formation are not well understood (*Kang, Fung & Van den Berghe, 2021*). During intestinal embryonic development, neuron bodies and processes with varying degrees of differentiation form a neural plexus network, with developing neuron clusters destined to become the future ganglia. Mechanical forces, intracellular Ca2+ transients, and adhesive capacity may assemble ENCCs into enteric neuron clusters during uniform colonization and differentiation, ultimately forming the myenteric ganglia (*Hackett-Jones et al., 2011*; *Kang, Fung & Van den Berghe, 2021*; *McKinney & Kulesa, 2011*; *Rollo et al., 2015*). The mutually induced differentiation of ENCCs and intestinal mesenchymal cells may also contribute

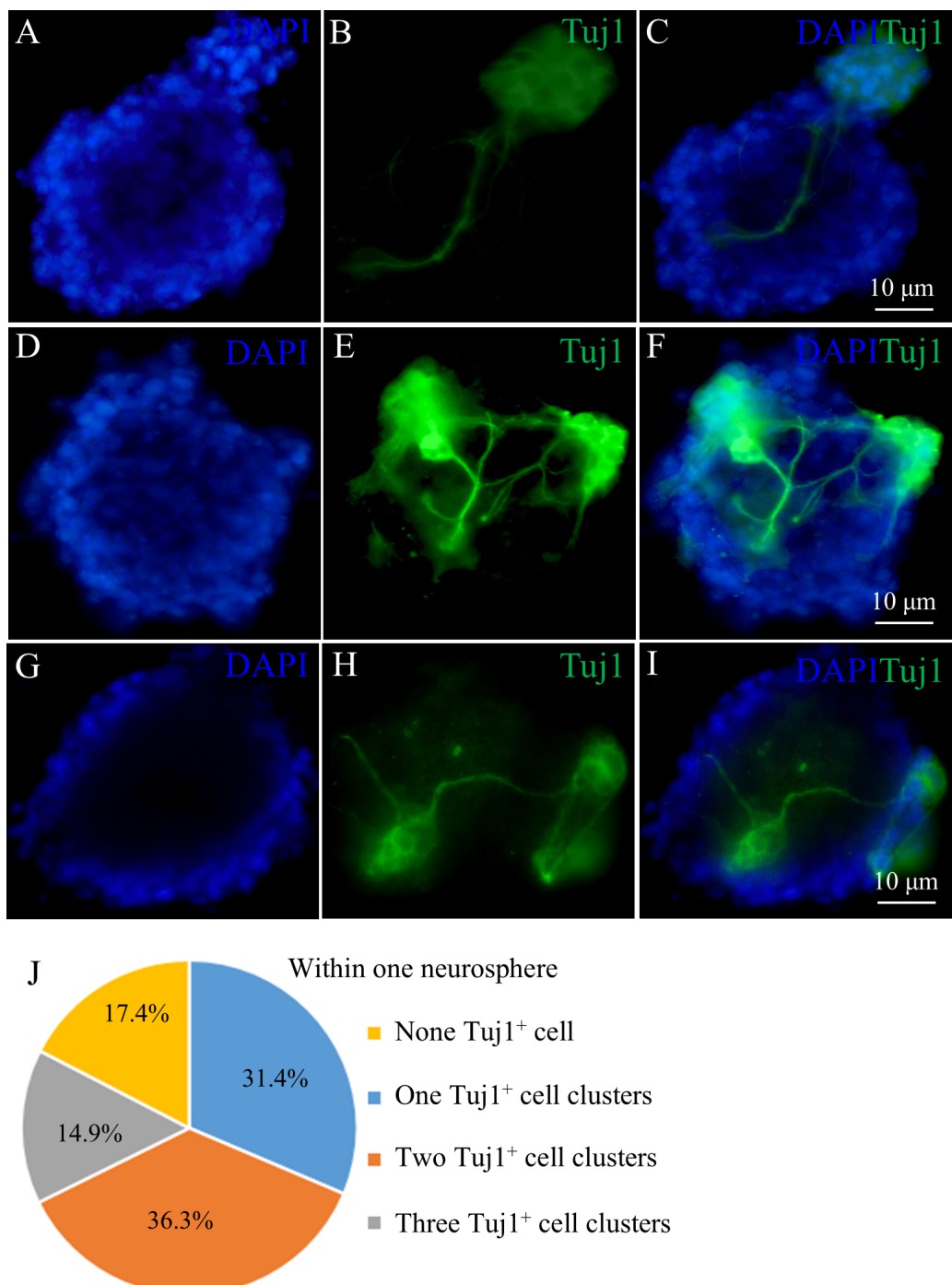

**Figure 6** **Varied neuronal cell clusters within neurosphere.** (A–C) Tuj1 labeling of a neuronal cell group within the neurosphere. (D–F) Tuj1 labeling of two distinct neuronal cell groups within neurosphere. (G–I) Tuj1 labeling of three separate neuronal cell groups within neurophere. (J) Percentage distribution of neuropheres containing varying numbers of neuronal cell clusters on the 6th day of culture. Three pregnant mice were used, thirty images were produced, and one set of data were generated.

to the formation of ganglia and smooth muscle cells (*Radenkovic, Radenkovic & Velickov, 2018*). In addition, the rapid expansion and stretching of the gut during development may lead to the formation of dispersed ENCC clusters, which eventually become ganglia and plexuses (*Chevalier et al., 2021*; *Xiao et al., 2022*). However, the factors determining the size of the myenteric ganglia and the number of intra-ganglionic neurons remain unknown.

We observed some intriguing phenomena during the *in vitro* culture of neurosphere. First, different clusters of neurons existed inside the neurosphere. In addition, individual neurons or small neurons clusters outside neurosphere might act as "bridge" to attribute to establishing neural networks (Fig. S7). In addition, retinoic acid was introduced into the cell culture on the 7th day to induce ENCCs or neuronal precursors to fully differentiate into neurons (*Li et al., 2023*). The neural network *in vitro* culture system showed some resemblance to the neural plexus network in the intestine (Fig. S8). Chen at al. suggested that enteric neurospheres within neural networks were similar to the enteric ganglia within myenteric plexus (*Chen et al., 2023*). *Nestor-Kalinoski et al. (2022)* reported that small triangular ganglia form the nodes of the network in the gut, constituting the connections between the ganglia. Therefore, we believe that there are similar classification phenomena between the neurospheres in the culture system and the ganglia in the myenteric plexus, but the specific mechanism needs to be further explored in the future.

# CONCLUSIONS

In conclusion, three distinct groups of myenteric ganglia in mice and humans were identified based on ganglionic area and neuron count, suggesting the presence of different classes from early developmental stages. These findings improve our understanding of ENS biology and may guide therapeutic approaches for Hirschsprung's disease and related disorders characterized by abnormalities in neuron count and plexus morphology.

# ACKNOWLEDGEMENTS

We are grateful that immunofluorescence experiments and flow cytometry assay were completed in Experimental Medical Research Center of Tongji Hospital, Tongji Medical College, Huazhong University of Science and Technology.

## Funding
This research was funded by National Natural Science Foundation of China, grant number 82071685, 82371720, and 82402021, China Postdoctoral Science Foundation, grant number 2024M751025, Hubei Provincial Key Research and Development Program, grant number 2023BCB095, and Natural Science Foundation of Fujian Province, grant number 2023J011306. There was no additional external funding received for this study. The funders had no role in study design, data collection and analysis, decision to publish, or preparation of the manuscript.

## Grant Disclosures

The following grant information was disclosed by the authors:

National Natural Science Foundation of China: 82071685, 82371720, 82402021.

China Postdoctoral Science Foundation: 2024M751025.

Hubei Provincial Key Research and Development Program: 2023BCB095.

Natural Science Foundation of Fujian Province: 2023J011306.

## Competing Interests

The authors declare there are no competing interests.

## Author Contributions

- Luyao Wu conceived and designed the experiments, performed the experiments, prepared figures and/or tables, authored or reviewed drafts of the article, and approved the final draft.
- Lei Xiang performed the experiments, authored or reviewed drafts of the article, and approved the final draft.
- Yingjian Chen conceived and designed the experiments, performed the experiments, prepared figures and/or tables, and approved the final draft.
- Handan Mao analyzed the data, prepared figures and/or tables, and approved the final draft.
- Xinyao Meng conceived and designed the experiments, analyzed the data, prepared figures and/or tables, and approved the final draft.
- Jing Wang analyzed the data, prepared figures and/or tables, and approved the final draft.
- Honglin Li analyzed the data, authored or reviewed drafts of the article, and approved the final draft.
- Xuyong Chen analyzed the data, authored or reviewed drafts of the article, and approved the final draft.
- Jiexiong Feng conceived and designed the experiments, authored or reviewed drafts of the article, and approved the final draft.
- Jun Xiao conceived and designed the experiments, performed the experiments, prepared figures and/or tables, authored or reviewed drafts of the article, and approved the final draft.

## Human Ethics

The following information was supplied relating to ethical approvals (*i.e.*, approving body and any reference numbers):

The Ethics Committee of Tongji Medical College, Huazhong University of Science and Technology (Approval number: 2021-S033).

## Animal Ethics

The following information was supplied relating to ethical approvals (i.e., approving body and any reference numbers):

The Institutional Animal Care and Use Committee of Tongji Medical College, Huazhong University of Science and Technology (Approval number: 2019-S2500).

## Data Availability

The raw measurements are available in the Supplementary Files.

## Supplemental Information

Supplemental information for this article can be found online at http://dx.doi.org/10.7717/peerj.19329#supplemental-information.

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
