# Peer review of "Three distinct classes of myenteric ganglia in mice and humans: insights from quantitative analyses"

_PeerJ, doi:10.7717/peerj.19329_

## Round 0.1 · original submission · Major Revisions

Dear authors,

Some important questions have been raised by the reviewers. Please, refer to their comments for further detail.

Also, please note, that any suggested bibliography are not mandatory addictions, particularly if the papers indicated do not make much sense in to the scope of your work.

·

Basic reporting

The report is easy to read and grammatically correct for the most part. The figures are clear, and the images are mostly of high resolution. The authors have also provided raw data tables which is appreciated.
1. The introduction needs to acknowledge more previous literature in the field. There are existing studies that have investigated myenteric ganglia morphology to a degree in both animal models and humans. This will help to more specifically describe the knowledge gap through lines 62-69. See the following articles for some examples:
a. Wattchow et al (Gastroenterology. 1995 Sep;109(3):866-75.)
b. Iwase et al (Journal of clinical gastroenterology, 2005; 39(8), 674-678.)
c. Ippolito et al (Cell Tissue Res 336, 191–201 (2009))
d. Gabella et al. (J Neurocytol 13, 49–71 (1984))
e. Genov et al (Acta Morphologica et Anthropologica 30: 1-2 (2023))
2. The images of human colon showcased in Figure 3A, B, E and F are partially out of focus or contain many bubbles. If possible, the authors should update these figures with in-focus images.
3. The claim that there is a lack of morphological studies characterising the myenteric plexus could be re-worded (Lines 24-25). (Refer to point #2 above).

Experimental design

The methodology is generally reported well throughout the manuscript. Catalogue numbers, reagent concentrations, and protocols are mostly included and are clear to follow.
1. If the age range of the human colonic tissue collected is known, then it would be beneficial to include (lines 114-116). Particularly since this manuscript is discussing developmental changes.
2. Specific antibody concentrations used should be reported in lines 172-174.
3. The frequency of media changes for the neurosphere cultures should be reported in lines 147-155.
4. Additional microscope settings (line 125) would be beneficial to include such as:
a. The model/brand of the confocal microscope used
b. Information of the objective used (magnification, numerical aperture)

Validity of the findings

The findings discussed are consistent with the results shown throughout the figures. The implications and conclusions are logical, and the authors do not overclaim the findings of the study.
1. Additional clarification to explain the differences between the results described in lines 196-226 would be beneficial. It is not completely clear how the results in lines 215-226 are different to results of 196-213. Some of this section sounds reiterative.
2. A sentence to describe the author’s rationale for the approach would be very helpful at the beginning of lines 244, and 271.
3. It is not clear how Figure S3 demonstrates resemblance of an intestinal neural plexus (line 317). Instead, the statement should be adjusted to: “shows some resemblance to the neural plexus…”. As this is more fitting to the supplemental figure provided.
4. The statistical tests used are logical, however the p-values for significant results are not specified.
5. Clearer reporting of the number of biological replicates would be helpful (eg. n=3).
6. The statement in line 294-295 does not make sense and should be re-written.

Additional comments

1. Rename “TUJ1 to Tuj1” for consistency throughout the manuscript (Line 123, 231 and Figure 3 legend).
2. Rename “four-week old” to “4-week old” for consistency throughout the manuscript (line 76).
3. Change “CoraL-ite” to “CoraLite” to match how it is spelled by the manufacturer (lines 101, 103, 141).
4. Remove “rigorous” from line 180.
5. The authors could state that HuC/D also labels the neuron cell body, not just the nucleus (line 190). This is already mentioned later in the manuscript (line 232).
6. The legend of asterisk p-value significance should be included in the Figure 4 legend.
7. There are typos in:
a. Line 153: “strep-tomycin” to “streptomycin”
b. Line 173: “an-ti” to “anti”
c. Line 185: “se-lection” to “selection”

Reviewer 2 ·

Basic reporting

no comment

Experimental design

Important to note that the data generated was not sex-segregated. This is a missed opportunity as studying sex-specific differences in the statistical model of ENS structure would be very important.

Validity of the findings

It is unclear how the authors have defined a ganglion to contain a single neuron. It is understandable to have two or three neurons or more to constitute a ganglion, but a single neuron is a single neuron and not a ganglion. Do the authors report that there are ganglia that are predominantly all Sox10+ cells as they contain multiple Sox10+ cells but only contain a single neuron? If so, please clarify.
In the same vein, are there ganglia that do not contain any neurons?

Additional comments

1. The title of the manuscript could be more detailed and precise.
2. It is important to discuss the relevance of differently sized ganglia to their component neurons. Would the three classes of ganglia contain different proportions of neuronal subtypes?
3. An important paper missing in discussion is Kobayashi et al eNeuro who showed that smaller ganglia are more frequent in the adult murine small intestine than larger ganglia. It will be important to see similar frequency distribution of the data generated by the authors to test if the patterns of relative abundance holds.
4. For the readers, it would help with reproducibility if the authors mark the various sizes of ganglia in the results figure section.
5. Figure-3 legend (page-20) indicates that in your study used 2 normal human samples and 3 hypoganglionosis patients, can the authors comment on whether these are age matched, and whether ganglia metrics differed between health and diseased condition.
6. According to lines 222-226, the myenteric ganglia of a 4-week-old mouse were classified into three types (such as having more neurons in the large ganglia and fewer in the small ganglia). Could this be because the ganglia are small due to a low number of neurons, or vice versa?

---

## Round 0.2 · accepted · Accept

Dear authors,

I am now accepting your manuscript for publication. Many thanks for your submission and collaboration. Congratulations!

Reviewer 2 ·

Basic reporting

No comment

Experimental design

No comment

Validity of the findings

No comment

Additional comments

The reviewer is thankful to the authors for their thoughtful response and for their excellent work and has no further comments